# Management of *Pyrenophora teres* f. *teres*, the Causal Agent of Net Form Net Blotch of Barley, in A Two-Year Field Experiment in Central Italy

**DOI:** 10.3390/pathogens11030291

**Published:** 2022-02-24

**Authors:** Francesco Tini, Lorenzo Covarelli, Giacomo Ricci, Emilio Balducci, Maurizio Orfei, Giovanni Beccari

**Affiliations:** Department of Agricultural, Food and Environmental Sciences, University of Perugia, 06121 Perugia, Italy; francesco.tini@collaboratori.unipg.it (F.T.); riccigiacomo7@gmail.com (G.R.); emilio.balducci@studenti.unipg.it (E.B.); maurizio.orfei@unipg.it (M.O.); giovanni.beccari@unipg.it (G.B.)

**Keywords:** *Pyrenophora teres* f. *teres*, net blotch, barley, fungicides, field management

## Abstract

*Pyrenophora teres* is the causal agent of barley net blotch (NB), a disease that can be found in two different forms: net form (NFNB), caused by *P. teres* f. *teres*, and spot form (SFNB), caused by *P. teres* f. *maculata*. A two-year field experiment was carried out to evaluate the response to NB of six different barley cultivars for malt or feed/food production. In addition, the efficacy of several recently developed foliar fungicides with different modes of action (SDHI, DMI, and QoI) towards the disease was examined. After NB leaf symptom evaluation, the identification of *P. teres* forms was performed. Grain yield was determined, and pathogen biomass was quantified in the grain by qPCR. In the two experimental years characterized by different climatic conditions, only *P. teres* f. *teres* was detected. The tested cultivars showed different levels of NFNB susceptibility. In particular, the two-row cultivars for malt production showed the highest disease incidence. All applied fungicides exhibited a high efficacy in reducing disease symptoms on leaves and pathogen accumulation in grains. In fact, high levels of fungal biomass were detected only in the grain of the untreated malting barley cultivars. For some cultivars, grain yield was positively influenced by the application of fungicides.

## 1. Introduction

Barley (*Hordeum vulgare* L.) is one of the most important cereal crops worldwide, with a cultivated surface of 51 million hectares and a production of about 159 million tons in 2019 [1]. Several fungal pathogens can seriously affect barley development, compromising final yield. One of the most devastating barley fungal diseases is net blotch, which may cause high yield losses and complete crop failure [2]. Net blotch is an extremely widespread disease occurring in all barley growing regions [3].

Originally, the causal agent of barley net blotch was named *Helminthosporium teres* (Sacc.) in 1809 and renamed as *Pyrenophora teres* Drechs. (anamorph *Drechslera teres* (Sacc.) Shoemaker) in 1930 [4]. *P. teres* can occur in two different forms: *P. teres* f. *teres* and *P. teres* f. *maculata*. The two forms are morphologically similar and are distinguished by the symptoms they cause on leaves. In detail, *P. teres* f. *teres* causes net form net blotch (NFNB), characterized by stretched and dark-brown lesions. Necrosis develops along leaf veins with occasional transverse striations forming a net-like pattern. *P. teres* f. *maculata* causes spot form net blotch (SFNB), with ovoid lesions surrounded by a chlorotic area. Less virulent strains of both forms can produce smaller necrotic lesions or lesions without surrounding chlorosis [5]. The two forms of the pathogen causing net blotch can be classified as separate based on genetic analysis [6,7,8,9,10,11]. The two forms have diverged from each other about 519 kya (±30) and from one of their closest relatives *P. tritici-repentis*, the causal agent of wheat tan spot, about 8.04 Mya (±138 kya) [9,12]. The symptoms caused on leaves by both forms of *P. teres* are partially induced by various phytotoxins [13], such as pyrenolides (A, B, C, and D), pyrenolines (A and B), and three peptide alkaloids—aspergilomarasmine A and its derivatives [14].

The integrated management of net blotch is primarily based on the adoption of correct agricultural practices. Crops residues, seeds, and wild grass species are the inoculum sources responsible for spreading the disease [15]. Primary inoculum destruction, sowing healthy seeds, crop rotation, and soil tillage are the first steps to effectively manage the disease [16]. A minimum of two years between barley crops is required to prevent net blotch [17]. Currently, reduced tillage, conservation tillage, or no-tillage cultivation methods are widespread, however, these practices reduce costs but simultaneously increase *P. teres* severity [18,19]. These factors, associated with short rotations, make the management of net blotch more challenging [20]. 

These considerations have led to an increased interest in the use of resistant cultivars for disease management [21] and several studies have been realized to identify resistant cultivars from the beginning of the last century [22]. Because of the high genetic variability observed in *P. teres*, the obtainment of barley lines with complete resistance to all pathogen isolates is very difficult [23]. However, genes for net blotch resistance or quantitative trait loci (QTL) have been identified and they are distributed on all seven barley chromosomes [15]. Nevertheless, to date, very few cultivars exhibit resistance to the net form of the disease, while most of them have a lower susceptibility reaction to the spot form [21]. Resistance to *P. teres* is a major priority of all barley breeding programs [24]. Due to the lack of completely resistant cultivars, in addition to correct agricultural practices, fungicide applications are largely used to control net blotch [25]. As mentioned previously, barley seeds are considered a very important source of inoculum, and net blotch is reduced when a fungicide seed dressing treatment is applied [26]. In addition, different groups of foliar fungicides are used for the control of this disease: succinate dehydrogenase inhibitors (SDHIs), demethylation inhibitors (DMIs), and quinone-outside inhibitors (QoIs) [27,28,29]. In particular, in many barley growing areas, QoIs show excellent efficacy and are registered for the control of net blotch [30] with SDHI and DMI fungicides also largely used in Europe [28] and worldwide [31]. The effectiveness of disease control using foliar fungicides depends on different factors, such as disease pressure, application rate, mode of action of the active ingredient, timing and number of treatments, and application rate [29]. When correctly applied to susceptible cultivars, fungicides can provide reduction of disease impact and may control other foliar fungal diseases of barley such as scald (caused by *Rhynchosporium secalis*), Ramularia leaf spot (caused by *Ramularia collo-cygni*), leaf rust (caused by *Puccinia hordei*), and powdery mildew (caused by *Blumeria graminis* f. sp. *hordei*). Currently, fungicides show high efficacy in net blotch control, but continuous and repeated applications may select fungal populations with fungicide resistance [27,32]. 

On the basis of these considerations, the present study aimed to explore the efficacy of an integrated control strategy towards barley net blotch. In detail, we show the results of a two year field experiment carried out in an area of central Italy under natural disease pressure. Preliminarily, the identification of the net blotch causal agent, naturally detected in the field during the survey, was carried out by isolation, morphological observations, and molecular analysis. Successively, the response of six commercially interesting barley cultivars (four two-row and two six-row cultivars) and the efficacy of the most recent foliar fungicides including different modes of action (QoI, SDHI, and DMI) were evaluated towards net blotch disease. The obtained results can help to develop and improve an integrated management strategy of this disease with the adoption of a correct choice of barley cultivars and a rational/sustainable use of fungicides.

## 2. Results

### 2.1. P. teres Identification

Symptoms observed on barley leaves were characterized by stretched and dark-brown lesions that developed along leaf veins with transverse striations, forming a net-like pattern. This manifestation resembled that of NFNB (Figure 1). 

Setting up of humid chambers allowed the evasion of pathogen reproductive structures from the symptomatic leaves and stereomicroscope (SZX9, Olympus, Tokyo, Japan) and optical microscope (Axiophot, Zeiss, Oberkochen, Germany) observations made their characteristics appreciable (Figure 1). Conidia were observed at the top of conidiophores swollen at the base and were present singly or in groups of two or three. Conidia were smooth, cylindrical, and straight; round at both ends; sub hyaline; and with four to six pseudosepta. On the basis of the morphological descriptions [5], we found that these characteristics resembled those of *P. teres*.

Simultaneously, from the leaves placed in humid chambers, we realized pathogen isolation (see Section 4.2 for details) and after amplification of the *Internal Transcribed Spacer* (ITS) region from the extracted DNA originating from a monoconidial colony developed on PDA, using primer ITS1 and ITS4, we identified the isolated pathogen, by BLAST analysis, as *P. teres*. 

In addition, the PCR protocol carried out using PttQ4F-PttQ4R and PtmQ10F-PtmQ10R primers showed that the fungal pathogen isolated from the symptomatic leaves was *P. teres* f. *teres*, the causal agent of NFNB. 

### 2.2. Weather Conditions Recorded in the Two Experimental Years

Weather data (temperature, rainfall, and relative humidity) recorded in the experimental field during the two investigated years are detailed in Figure 2. In particular, climatic data are shown from 1 February (approximately stage BBCH 23 of barley cultivars) to 4 July (BBCH 99), both for 2019 and 2020 (Figure 2).

In general, low rainfall levels (average and total amounts) were observed in 2019 (2.0 mm and 309.8 mm, respectively) and 2020 (1.3 mm and 193.8 mm, respectively), but their distribution was very different in the two experimental years. 

In particular, considering the climatic requirements of the main barley fungal pathogens and the period during which foliar fungal diseases of barley may occur in central Italy, we considered a time frame of 70 days from 8 March (6th week, BBCH 30) to 16 May (15th week, BBCH 70). During the first year (2019), the daily average and total rainfall amounts recorded in the above-mentioned period were 2.28 mm (daily average) and 159.4 mm (total), whereas they were 0.52 mm (daily average) and 36.4 mm (total) during the second year (2020). 

During the entire weather recording time (from 1 February to 4 July), the two years were similar for relative humidity (average levels of 71.2% in 2019 and 66.7% in 2020), but, again, focusing on the period from 8 March to 16 May, we recorded an important difference between the two years. In fact, reflecting the rainfall levels previously mentioned, the average level of relative humidity was 72.5% during the first year (2019), whereas it was 63.8% during the second year (2020). 

Concerning the whole recording time, the two investigated years did not show any particular differences in terms of temperature, with averages of 16 °C in 2019 and 15.7 °C in 2020. Moreover, considering the same period from 8 March to 16 May, we found that only small differences were recorded in terms of average temperature between the two years (12.5 °C in 2019 and 13.1 °C in 2020).

In conclusion, with particular regard to rainfall and relative humidity, the second experimental year (2020) was characterized by a drier spring that might have disadvantaged the development of fungal leaf diseases of barley, such as net blotch, with respect to the first year. 

### 2.3. NFNB Visual Score and Production Parameters

Data relative to the presence of net blotch symptoms in both experimental years (2019 and 2020) are shown in Figure 3. Symptoms on barley leaves were visually estimated 30 days after fungicide application (BBCH 80), before senescence of the main leaves (see Section 4.1). 

During the first experimental year (2019; Figure 3a), the three different fungicides (Aviator Xpro, Elatus Era, Priaxor) applied at BBCH39 significantly reduced (*p* ≤ 0.05) NFNB symptoms on each barley cultivar in comparison to the untreated control. On average, the application of fungicides reduced the disease index by 78%. No significant differences (*p* ≥ 0.05) were recorded in terms of symptom reduction between the three different fungicides in all tested cultivars. Focusing on the untreated controls only, we found significant differences (*p* ≤ 0.05) in terms of net blotch susceptibility among the six tested cultivars. In detail, the susceptibility trend was Sunshine > Quench = Ketos > Alimini = Capricorn = Atomo. 

As mentioned previously (Section 2.2), the second experimental year (2020) was characterized by lower total spring rainfalls (mm), in comparison to the first year (2019). This influenced net blotch development causing a generally low disease level. In fact, when comparing net blotch symptoms observed on the untreated controls only (Appendix A), we detected a higher disease index in 2019 than in 2020 for Alimini (*p* = 0.027), Atomo (*p* = 0.012), Capricorn (*p* = 0.012), Quench (*p* = 0.003), and Sunshine (*p* = 0.00001) cultivars. We did not observe any significant symptom differences (*p* = 0.058) between the two experimental years only in the cultivar Ketos. 

In the epidemiological conditions that occurred in 2020, all fungicides applied at BBCH39 completely reduced (*p* ≤ 0.05) symptom development on each tested barley cultivar (Figure 3b). In general, fungicide applications reduced NFNB symptoms by 99.8% with no significant differences (*p* ≥ 0.05) recorded in terms of symptom reduction following the application of the three different fungicides. Moreover, for this experimental year, on the basis of the disease index detected only in the untreated controls, the susceptibility of the tested barley cultivars followed a trend similar to that recorded in the first year: Sunshine ≥ Quench ≥ Ketos > Alimini = Capricorn = Atomo.

Data relative to production parameters, such as grain yield (t ha^−1^), protein content (%), and kernel specific weight (kg hL^−1^), for both experimental years, are summarized in Table 1. 

During the first year (2019), the application of the three fungicides significantly increased (*p* ≤ 0.05) total yield (t ha^−1^) in comparison to the untreated control (Table 1) only for the cultivars Ketos, Quench, and Capricorn. In Alimini, Atomo, and Sunshine, no significant differences (*p* ≥ 0.05) were detected between the untreated control and fungicide treatments. In the same year (2019), the protein content (%) did not show any change (*p* ≥ 0.05) following fungicide applications (Table 1). Similarly, the kernel specific weight (kg hL^−1^) was unaffected (*p* ≥ 0.05) by fungicide treatments in Alimini, Capricorn, and Atomo cultivars (Table 1), whereas it significantly (*p* ≤ 0.05) increased in Sunshine and Quench.

### 2.4. P. teres f. teres Biomass in Barley Grains

A preliminary PCR analysis, performed with species-specific primers, showed the presence of *P. teres* f. *teres* in the harvested grains (data not shown). 

Data relative to biomass quantification of *P. teres* f. *teres* in barley grains in the two experimental years (2019 and 2020) are shown in Figure 4.

Real-time qPCR (qPCR) was used to quantify pathogen biomass in the grains of the six experimented barley cultivars. The reaction efficiency calculated from *P. teres* f. *teres* linear equation was 103%. The R^2^ value calculated from the linear equation of the *P. teres* f. *teres* standard curve was 0.999. The dissociation curve analysis showed specific amplification products in the presence of pure fungal DNA (standard curves) and in the presence of *P. teres* f. *teres* DNA (samples). No target amplification was detected in the negative controls. Therefore, the Ct values used to quantify fungal biomass were those for which dissociation curve analysis showed the presence of specific amplification products. The reaction efficiency calculated from the barley linear equation was 100% with R^2^ = 0.998.

In general, *P. teres* f. *teres* biomass was detected (Figure 4a,b) by qPCR only in the grains of Sunshine and Quench cultivars in both experimental years. Conversely, in Atomo, Alimini, Ketos, and Capricorn, pathogen accumulation in grains was under the limit of detection (LOD) in both years. In detail, *P. teres* f. *teres* accumulation in Quench and Sunshine grains was significantly reduced following fungicide application at BBCH 39. 

Considering the untreated controls only, we found that the pathogen biomass present in the grains of Sunshine and Quench cultivars was not significantly different in both years (*p* = 0.10 in 2019 and *p* = 0.65 in 2020, respectively). Comparing *P. teres* f. *teres* accumulation in Quench grains (untreated controls) harvested in 2019 with respect to 2020, we detected a significant difference (*p* = 0.02) with a higher biomass content in the grains coming from the first experimental year. Conversely, in the cultivar Sunshine, no difference between pathogen biomass in the grains harvested in the two experimental years was detected (*p* = 0.67).

### 2.5. Relationship between P. teres f. teres Biomass in the Grains and NFNB Symptoms on Leaves

Upon finding that *P. teres* f. *teres* was only detected in the grains of Sunshine and Quench cultivars, we analyzed the relationship between leaf symptoms (%) and pathogen accumulation in the grains (pg *P. teres* f. *teres* DNA ng barley DNA^−1^) (Figure 5).

Considering both experimental years and the two malting barley cultivars (Quench and Sunshine) together, we found that *P. teres* f. *teres* accumulation in grains showed a positive and significant correlation (r = 0.70; *p* = 0.000001) with NFNB symptoms observed at BBCH 80 (30 days after fungicide application). 

Moreover, considering the experimental years singularly and Quench and Sunshine cultivars together, the pathogen biomass showed a positive and significant correlation, with NFNB symptoms observed both in 2019 (r = 0.86, *p* = 0.00001) and in 2020 (r = 0.90; *p* = 0.012).

Focusing on the single cultivars, we observed a positive correlation in both experimental years. However, the relationship was significant only in 2019. 

In detail, a significant relationship (r = 0.94, *p* = 0.003) was observed in the Quench cultivar in 2019. For the same cultivar, a positive, but not significant (r = 0.93, *p* = 0.082), correlation was detected in 2020. In 2019, a positive and significant relationship (r = 0.83, *p* = 0.001) was observed in the Sunshine cultivar. Conversely, for the same cultivar, a positive, but not significant (r = 0.97 and *p* = 0.082), correlation was observed in 2020.

Concerning the other tested cultivars, no correlations were assessed due to the absence of pathogen DNA in their grain (<LOD).

## 3. Discussion

Net blotch, caused by *P. teres* f. *teres* (net form, NFNB) and by *P. teres* f. *maculata* (spot form, SFNB), is a worldwide barley foliar disease. It can cause yield losses [33,34] as well as grain quality reduction [15,35,36,37]. 

Due to the global impact of this disease, the present study aimed at exploring the management strategies for controlling *P. teres* f. *teres* by comparing six different barley cultivars and three recent fungicides available in the Italian market.

The experiment was carried out during two barley growing seasons (2018/2019 and 2019/2020) under natural inoculum pressure. The first step of this study was the identification of the fungal pathogens colonizing barley leaves following the detection of specific symptoms. Visual observation of symptoms, pathogen isolation, morphological characteristic of reproductive structures, and molecular analysis led to the identification of *P. teres* f. *teres* as the pathogen responsible for the observed symptomatology. No *P. teres* f. *maculata* was detected. The occurrence of only one form of the pathogen in a given field has also been described in the past [38]. In addition, one form of net blotch is often dominant within a barley-growing region [2,39,40]. In fact, for example, *P. teres* f. *maculata* has become more prevalent in North Dakota (USA), Idaho (USA), Victoria (Australia), Turkey, and the western provinces of Algeria [41,42,43,44,45]. Conversely, *P. teres* f. *teres* is the most present form in northeastern Algeria, East Azerbaijan, Finland, and in most parts of the Ethiopian highlands [46,47,48,49].

*P. teres* is a hemibiotrophic pathogen that survives saprophytically between cropping seasons. Its mycelium can be present in the seeds before sowing, on wild grasses, and on host crop residues, forming a source of primary inoculum [50,51]. Favorable environmental conditions such as prolonged wet periods, increase primary inoculum levels [52]. In fact, low rainfall can provide a relatively low net blotch incidence [52]. 

In this work, the key period in which barley foliar diseases usually occur was characterized by low rainfalls during the first year (2019) and by dry conditions during the second one (2020). Nevertheless, in the two surveyed years, symptoms attributable to NFNB were observed on the leaves of the six barley cultivars included in the experiment, even if with different incidence levels between the two years. In general, higher NFNB symptoms were observed in 2019 compared to 2020, probably due to the different climatic conditions that occurred in the two experimental years. However, in both years, the same trend of cultivar susceptibility to NFNB was observed. In detail, the two-row cultivar Sunshine showed the highest incidence of NFNB symptoms, followed by Quench (both two-row cultivars for malt production). The highest susceptibility to NFNB of the two-row barley cultivars in comparison to six-row ones has been previously observed [53]. Moreover, higher resistance levels to NBSF of six-row barley cultivars compared with two-row barley cultivars have been reported [54]. However, in the present study, the two-row cultivars Atomo and Capricorn (for feed use) showed the lowest NFNB incidence. This is in accordance with a study [55], in which about 40% to 65% of lines from two-row barley Australian populations had a resistant reaction to NBSF. The results of the present study show a higher susceptibility to NFNB of the barley cultivars used for malt obtainment. Therefore, this disease could be a risk for the malting industry, which is in constant expansion. Currently, malt producers have a limited number of cultivars available for malt production, and their high susceptibility to net blotch may represent a limiting factor for their cultivation in certain areas. 

Comparing the results of the two experimental years, we found that the amount of pathogen DNA detected in the grain of the Sunshine cultivar was very similar. Conversely, the fungal biomass level detected in the grain of the Quench cultivar increased in the second year. This may be explained by the interaction between the plant genotype and weather conditions. It could be possible that in dry conditions, Quench became more susceptible to the disease, allowing a higher pathogen translocation from leaves to grain. Conversely, the absence of pathogen detection by qPCR in the six-row and two-row cultivars for feed use may have been due to the lower susceptibility of these genotypes compared to the malting ones. In less susceptible genotypes, some host defense mechanisms could have prevented pathogen translocation from the leaves to the grain, making it not detectable by qPCR. These mechanisms shall be further investigated in future studies.

This study showed also that all tested fungicides (Aviator Xpro—prothioconazole + bixafen; Elatus Era—benzovindiflupyr + prothioconazole; Priaxor—fluxapyroxad + pyraclostrobin) showed a high efficacy in controlling net blotch. In general, in 2019, the three fungicides considerably reduced (–78%) the observed symptoms with respect to the untreated controls. In 2020, a year characterized by a lower symptom incidence, the three fungicides reduced completely (–99.8%) NFNB disease index in comparison to the untreated controls. Some of these active ingredients are well known to be very effective also when used singularly. In fact, even if in this study a mixture of fluxapyroxad and pyraclostrobin (Priaxor) was tested, fluxapyroxad (an SDHI fungicide) alone was recently defined as the best *P. teres* inhibitor [31]. Similarly, the very high efficacy of pyraclostrobin (QoI) alone towards this pathogen has been previously demonstrated [30,56]. Conversely, the only use of DMIs may not be highly efficient in controlling this disease [56]. In addition, in net blotch management, the use of mixtures of fungicides with different modes of action represents an important tool to prevent fungicide resistance [57,58,59,60,61].

In fact, fungal populations can evolve in response to fungicide pressure with the selection of different types of resistance mechanisms [62], such as the well-known sensitivity decline of the two forms of net blotch pathogens to triazole fungicides [27,56]. Furthermore, the selective pressure of a specific fungicide may favor a fungal species to the detriment of another one [63]. 

The positive effect of fungicide treatments on NFNB management was partially confirmed also by yield parameters. In fact, in the most susceptible cultivars, yield (t ha^−1^) increased following fungicide application at BBCH 39 in 2019 (Quench) and 2020 (Sunshine). Similarly, specific weight (kg hL^−1^) increased in 2019 for Quench and Sunshine cultivars. Fungicide applications on malting barley are known to be able to reduce leaf disease severity, increasing yield and kernel weight [64]. In addition, in the first experimental year, the cultivars Ketos (six-row for feed use) and Capricorn (two-row for feed use) showed a significant positive effect of fungicide application on final yield, showing the impact of treatments at BBCH 39 in foliar disease management in the season in which weather conditions were more favorable to disease development (2019). During 2020, with the exception of Sunshine cultivar, no differences between fungicide treatments were detected for yield and specific weight showing that, in the years in which weather conditions are unfavorable to net blotch development, fungicide application could also be avoided. Therefore, the monitoring of weather conditions plays a key role for the sustainable use of fungicides in the integrated management of this disease.

As *P. teres* f. *teres* is also a seed-borne pathogen, its biomass was quantified by qPCR in the harvested grains. The pathogen accumulated only in Sunshine and Quench cultivars, both in 2019 and in 2020, and foliar fungicide applications at BBCH39 strongly reduced its translocation into the grains. This finding could be crucial for the obtainment of healthy barley seeds. Moreover, a positive relationship was observed between leaf symptoms in both experimental years and the amount of *P. teres* f. *teres* detected in the grains. Moreover, this result is particularly important for healthy seed reproduction because an estimation of barley grain infection with *P. teres* could be also predicted by observing leaf symptoms during the growing seasons.

## 4. Materials and Methods

### 4.1. Field Experimental Design and Fungicide Application

The study was realized on six barley cultivars: Alimini, Ketos, Capricorn, Atomo, Quench, and Sunshine, grown during the years 2018–2019 and 2019–2020 (named 2019 and 2020, respectively), in 1.5 m × 6 m (9 m^2^) experimental plots, in a randomized block design field trial. Details on the barley cultivars tested in the experiments are reported in Table 2.

The experimental field was located at the “FIELDLAB” of the Department of Agricultural, Food and Environmental Sciences of the University of Perugia (Papiano, Umbria, Italy, 42°57′ N, 12°22′ E, 165 m a.s.l.). Barley cultivars, seed-dressed with the fungicide Rancona 15 ME (Ipconazole 15 g L^−1^; UPL Agricultural Solutions Italy, Milan, Italy), were sown in the third week of November for the 2019 experiment and in the second week of December for the 2020 experiment. To promote the development of the main foliar barley diseases, the trial was performed on a field in which barley was the previous crop, in both experimental years. A nitrogen topdressing rate of 120 kg ha^−1^ with urea was applied after a basal fertilizer application of 75 kg P_2_O_5_ ha^−1^ in both years. Weeds were controlled with a post-emergence treatment (3 L ha^−1^ of Manta Gold, clopyralid 23.3 g L^−1^ + fluroxypyr 60 g L^−1^ + MCPA 266.7 g L^−1^; Syngenta Italia, Milan, Italy) + 1.25 L ha^−1^ of Columbus (clopyralid 80 g L^−1^ + fluroxypyr 144 g L^−1^ + florasulam 2.5 g L^−1^, Corteva Agriscience Italia, Milan, Italy) applied at the end-tillering stage (BBCH 29). During the flag leaf stage (flag leaf fully unrolled and the ligule just visible, BBCH 39), three different fungicides (Aviator Xpro—Bayer CropScience, Milan, Italy; Elatus Era—Syngenta Italia, Milan, Italy; Priaxor—BASF Italia, Milan, Italy; Table 3) were applied with a hand-pump sprayer equipped with 500 L ha^−1^ nozzles, using the active ingredient doses recommended on the labels. Each fungicide application was repeated on three plots (replicates) of the same cultivar. Three plots per cultivar were left untreated (controls), for a total of 12 plots per cultivar and a grand total of 72 plots (12 plots per cultivar x 6 tested cultivars) per year.

In addition, at the late booting stage (BBCH 45), an insecticide treatment was applied with Decis EVO (deltamethrin 25 g L^−1^, Bayer Crop Science).

All fungal diseases were evaluated under natural inoculum pressure. The experimental plots were regularly monitored throughout the growing seasons to assess the phytosanitary conditions of the tested materials. Symptoms on barley leaves were recorded after 30 days from fungicide application (BBCH 80), at the maximum symptom expression levels, before the beginning of leaf senescence, which could have negatively affected their correct scoring. The whole plots were visually scored by recording the percentage of plants showing NFNB symptoms in the two upper leaves (incidence %) and, focusing on symptomatic plants only, by scoring the average percentage of symptomatic area (severity %) in the same leaves. Foliar symptoms in the plots were finally expressed as a disease index (%) using the following formula:Foliar disease index (%) = [(incidence (%) × severity (%))/100].

At physiological maturity (BBCH 99), plots were harvested with a cereal plot harvester (Wintersteiger Italia Srl, Bozen, Italy), grains were separated from chaff material through a head thresher, and grain yield (t ha^−1^) was determined by adjusting kernel moisture content to 13%. In addition, specific weight (kg hL^−1^) and protein content (%) were also determined with a grain analyzer (Infratec 1241, Foss Headquarters, Hillerod, Denmark).

Furthermore, weather data (minimum, maximum and average temperatures, relative humidity, and rainfall) were recorded daily from sowing to harvest in both experimental years with a weather station located at the “FIELDLAB”. In detail, a time frame of the barley growing season during which climatic factors usually have the greatest influence on net blotch development in the experimental area [from 1 March (BBCH25) to 4 July (BBCH99) for a total of 18 weeks per year] was considered. The weekly average of weather data was then calculated.

### 4.2. Pathogen Isolation and Identification

During visual scoring of barley net blotch (see Section 4.1), symptom distribution in the tested cultivars, with different incidences and severities, was assessed. To perform the identification of the disease causal agents, we randomly collected 10 symptomatic leaves of the cultivars showing a high symptom expression (Sunshine and Quench). 

Leaves were then placed into humid chambers previously obtained by placing 40 mL of 1% Agar (Biolife Italiana, Milan, Italy) in Petri dishes (150 mm diameter; Nuova Aptaca, Canelli, Italy). After 24 h, leaves were subject to stereomicroscope (SZX9, Olympus) and microscope (Axiophot, Zeiss) observations to detect possible reproductive structures of the pathogen. Using a stereomicroscope, we placed a single conidium from the end of a conidiophore into Petri dishes (60 mm diameter; Nuova Aptaca) containing potato dextrose agar (PDA; Biolife Italiana) supplemented with streptomycin sulfate (0.16 g L^−1^, Sigma-Aldrich, St. Louis, MO, USA) with a thin needle under sterile conditions. The dishes were incubated in the dark at 22 °C for two weeks.

Molecular identification of isolated strains was realized by extracting DNA from the fungal colonies following the previously described method [65]. Briefly, mycelium was scraped from the surface of each colony with a micropipette tip and placed into a 200 µL plastic PCR tube (Thermo Fisher Scientific, Waltham, MA, USA) containing 50 µL of Extraction Solution (Sigma-Aldrich). Samples were homogenized for about 20 s with a micropipette tip, placed into a T-100 thermalcycler (Biorad, Hercules, CA, USA) at 99 °C for 10 min, and centrifuged at 12,470× *g* for 3 min with a 1–4 Sigma-Aldrich centrifuge. Fifty µL of dilution solution (Sigma-Aldrich) were added to the centrifuged samples, and 20 µL of the supernatant were transferred into a new 200 µL plastic tube (Thermo Fisher Scientific) and diluted with sterile water for molecular biology use (5prime, Hilden, Germany) to obtain a DNA concentration of ~30 ng µL^−1^. The DNA extracted was subject to partial ribosomal DNA (rDNA) *ITS* gene amplification with ITS1 and ITS4 primers (Appendix A) [66]. Each reaction contained 29 μL of sterile water for molecular biology use, 5 μL of 10X Dream Taq Buffer + magnesium chloride (Thermo Fisher Scientific), 3.75 μL of cresol red (Sigma Aldrich), 5 μL of dNTP mix 10 mM (Microtech, Naples, Italy), 2.5 μL of 10 μM ITS1 and ITS4 primers, 0.25 μL of 5 U μL^−1^ Dream Taq Polymerase (Thermo Fisher Scientific), and 2 μL of template DNA. The PCR consisted of an initial denaturation step (94 °C for 5 min), followed by 35 cycles of denaturation (94 °C for 30 s), annealing (57 °C for 30 s), extension (72 °C for 1 min), and of final extension (72 °C for 10 min). PCR fragments were visualized on TAE 1X agarose gel (2%) containing 500 μL L^−1^ of FireRed (Applied Biological Materials, Richmond, BC, Canada). DNA fragments were separated with an electrophoresis apparatus applying a tension of 110 V for ≈40 min. Electrophoretic runs were observed with an ultraviolet transilluminator (Uvitec Ltd., Cambridge, United Kingdom). The size of the amplified fragments was obtained by comparison with HyperLadder 100–1000 bp (Bioline Meridian Bioscience, Cincinnati, OH, USA). The obtained PCR fragments were purified and sequenced by an external sequencing service (Genewiz Genomics Europe, Takeley, United Kingdom). The sequences acquired were verified by Chromatogram Explorer Lite v4.0.0 (Heracle Biosoft srl 2011) and analyzed with the BLAST database [67].

On the basis of the BLAST analysis results, as well as with the support of the microscopic observations, we identified the isolated colonies as *P. teres*. Therefore, in order to identify the pathogen form, we performed a PCR analysis using PttQ4F-PttQ4R and PtmQ10F-PtmQ10R primers (Appendix A) [68]. The use of these primers allows for the distinguishing between *P. teres* f. *teres* and *P. teres* f. *maculata*. A PCR protocol was adopted as previously described. The PCR cycle consisted of an initial denaturation step (95 °C for 7 min), followed by 35 cycles of denaturation (95 °C for 30 s), annealing (60 °C or 62 °C (depending on primers) for 30 s), and extension (72 °C for 20 s), with a final extension (72 °C for 7 min). PCR fragments were visualized as previously described. The size of the amplified fragments was obtained by comparison with HyperLadder 100–1000 bp (Bioline Meridian Bioscience).

### 4.3. Pyrenophora teres f. teres Biomass Quantification in Barley Grains

DNA extraction from the 144 grain samples (72 samples from each experimental year) was obtained following the previously mentioned method [69] with some modifications described in [70]. DNA concentrations were determined with a Qubit^®^ 3.0 fluorometer (Thermo Fisher Scientific), using the dsDNA Broad Range (BR) Assay kit (Thermo Fisher Scientific), following the manufacturer’s protocol. Successively, the final concentration of each DNA sample was adjusted to 35 ng µL^−1^.

To define which form of *P. teres* was present in the sampled kernels, a preliminary study of the DNA extracted from grains by PCR was carried out using PttQ4F-PttQ4R and PtmQ10F-PtmQ10R (Appendix A) [69] with the protocol described in Section 4.2. 

On the basis of this result, as *P. teres* f. *teres* was the only *P. teres* form detected in the barley grains, its quantification following the previously described method [70] was realized. Briefly, DNA extractions from pure *P. teres* f. *teres* culture (Ptt-S strain of the fungal collection of the Department of Agricultural, Food and Environmental Sciences; University of Perugia, Italy) and barley grains (cv. Atomo) were preliminarily carried out to set up standard curves. A *P. teres* f. *teres* strain was grown on two PDA dishes for three weeks before DNA extraction. Fungal mycelium was scraped using a spatula, placed into 2 mL sterile plastic tubes (Eppendorf, Hamburg, Germany), freeze-dried with a Heto Powder Dry LL3000 (Thermo Fisher Scientific) lyophilizer, and finely ground with a Mixer Mill MM200 (Retsch, Haan, Germany). *P. teres* f. *teres* DNA was extracted as described in [71], whereas DNA from uninfected barley grains was extracted following the method described in [69], with slight modifications [70]. DNA concentrations were determined as previously described. A dilution series from 5 ng to 0.05 pg of *P. teres* f. *teres* DNA and from 50 ng to 5 pg of barley grains DNA, with a dilution factor of 10, were used to plot standard curves in each qPCR set. In each assay, two technical replicates were used. Standard curves were generated by plotting the logarithmic values of known DNA quantities in comparison with the corresponding cycle threshold (Ct) values. For each standard curve, from the average Ct of each dilution, the linear equation (y = mx + q), the R^2^ value, and the reaction efficiency [10(−1/m)] were calculated. The limit of detection (LOD) of *P. teres* f. *teres* DNA was 0.05 pg. qPCR analyses were carried out using species-specific primers for the quantification of *P. teres* f. *teres* (PttQ4F-PttQ4R; Appendix A), whereas *translation elongation factor 1α* (*tef1α*) primers (Appendix A) [72] were used for the quantification of barley grain DNA. To optimize qPCR reactions, annealing temperatures were experimentally adjusted. qPCR assays were carried out in a CFX96 real-time PCR detection system (Bio-Rad). The qPCR mix was composed of 2.5 µL of total DNA, 6 µL of SYBR^®^ Select Master Mix CFX (Thermo Fisher Scientific), 1.5 µL of 2 µM of each primer, and 0.5 µL of sterile water (5prime) for a total reaction volume of 12.5 µL. The qPCR program consisted of 50 °C for 2 min, 95 °C for 10 min, 45 cycles at 95 °C for 15 s, 61 °C for 1 min, heating at 95 °C for 10 s, cooling at 65 °C, and finally an increase to 95 °C of 0.5 °C every 5 s with the measurement of fluorescence. A dissociation curve was included at the end of the qPCR program to monitor the presence of potential primer-dimers and non-specific amplification products. Two analytical replicates for each sample were used in each assay. The fungal biomass in the barley grain was expressed as the ratio of *P. teres* f. *teres* DNA (pg) to barley DNA (ng).

### 4.4. Statistical Analysis

Data regarding foliar symptoms were subject to one-way ANOVA by considering each “experimental treatment” as the factor and “foliar symptoms (%)” as the variable. Furthermore, the six barley cultivars were tested for their susceptibility to NFNB by comparing the foliar symptoms observed on the control plots, considering the “control treatment” of the different cultivars as the factor and “foliar symptoms (%)” as the variable.

To investigate *P. teres* f. *teres* biomass accumulation in the grains of the six barley cultivars, we used one-way ANOVA. In detail, different “experimental treatments” were considered as the factor, whereas “fungal biomass” (expressed as pg of *P. teres* f. *teres* DNA ng^−1^ of barley DNA) was the variable.

Yield data were subject to one-way ANOVA by considering each “treatment” as the factor and “yield (t ha^−1^)” as the variable. The same approach was also used for protein content (%) and kernel specific weight (kg hL^−1^).

In all cases, one-way ANOVA and Duncan’s multiple comparison tests, to test pairwise contrasts (*p* ≤ 0.05), were performed with the Microsoft Excel Macro “DSAASTAT ver. 1.0192” [73]. Finally, the correlation between leaf symptoms (%) and *P. teres* f. *teres* biomass in the grains was analyzed using the Pearson correlation coefficient (r), followed by Student’s *t*-test.

## Figures and Tables

**Figure 1 pathogens-11-00291-f001:**
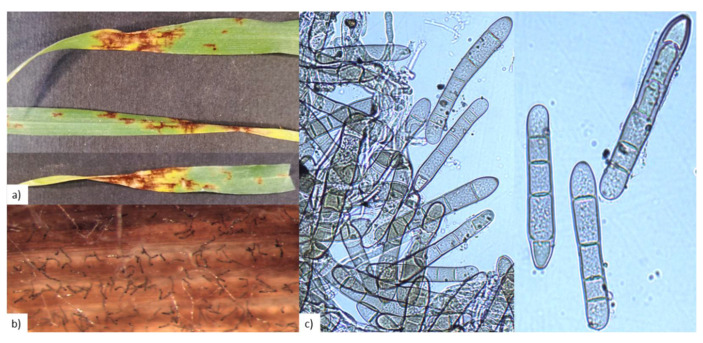
Symptoms on barley leaves (cv. Sunshine) characterized by stretched and dark-brown lesions that developed along leaf veins with transverse striations forming a net-like pattern (**a**). Conidiophores (**b**) and conidia (**c**) observed with stereomicroscope (SZX9, Olympus) and optical microscope (Axiophot, Zeiss), respectively. Conidia were observed at the top of conidiophores (**b**) and were smooth, cylindrical, and straight, round at both ends, subhyaline, and with four to six pseudosepta (**c**).

**Figure 2 pathogens-11-00291-f002:**
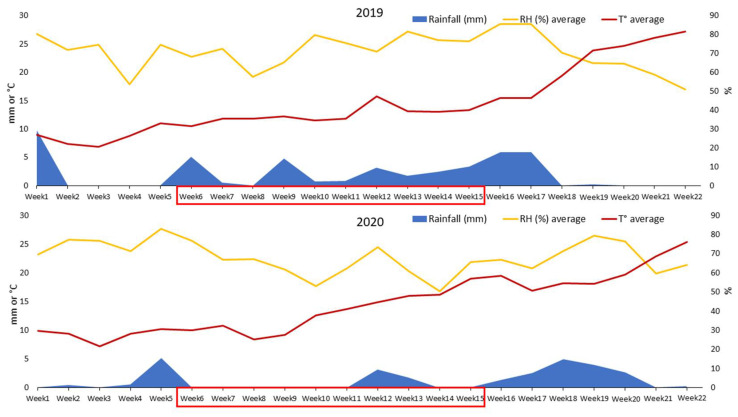
Weather data [rainfall (mm), relative humidity (%), and temperature (°C)] recorded at the Experimental Station “FIELDLAB” (Perugia, central Italy) where the field experiment was carried out in the two experimental years (2019 and 2020). For each experimental year, 18 weeks (from 1 February, BBCH 23 to 4 July, BBCH 99) were considered, and the average weekly data of rainfall (mm), relative humidity (%) and temperature are shown. The red square indicates the time frame (8 March, BBCH 30 to 16 May, BBCH 70) during which barley foliar diseases usually occur in central Italy.

**Figure 3 pathogens-11-00291-f003:**
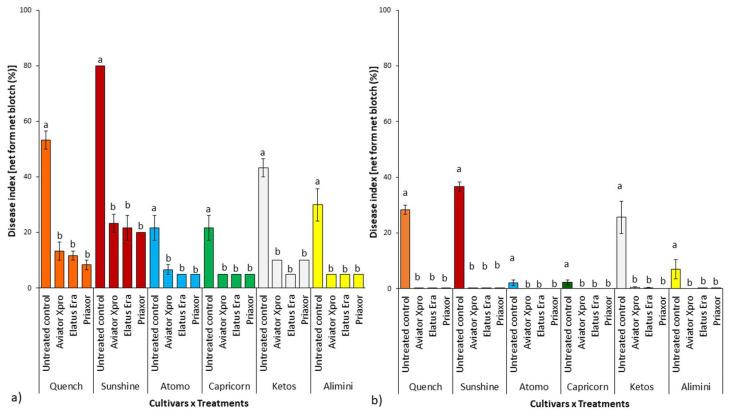
Average (three replicates) of disease index [net form net blotch symptoms (%)] observed during 2019 (**a**) and 2020 (**b**) on the six tested barley cultivars. Each cultivar was subject to three different fungicide applications and had one untreated control. Letters (a-b) above the columns indicate the presence, within each barley cultivar, of statistically significant differences (*p* ≤ 0.05) according to Duncan’s multiple comparison tests.

**Figure 4 pathogens-11-00291-f004:**
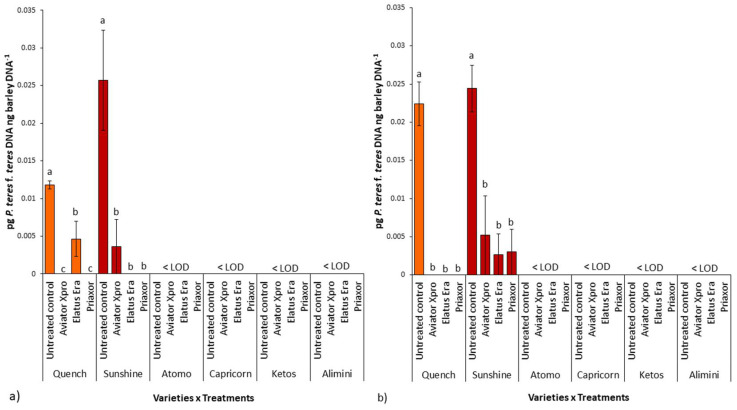
Average (three replicates) of *P. teres* f. *teres* biomass accumulation (pg DNA ng barley DNA^−1^) quantified by qPCR during 2019 (**a**) and 2020 (**b**) in the grains of the six barley cultivars tested. Each cultivar received three different fungicide applications and had one untreated control. Within each cultivar, letters (a-b) above the columns indicate the presence of statistically significant differences (*p* ≤ 0.05) according to Duncan’s multiple comparison tests. LOD is the limit of detection.

**Figure 5 pathogens-11-00291-f005:**
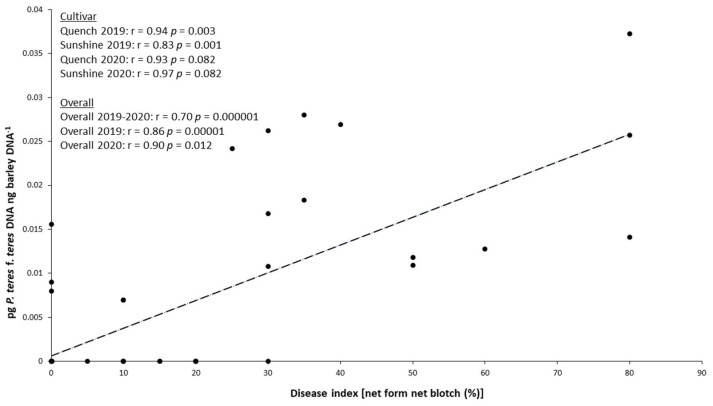
Correlation between *P. teres* f. *teres* biomass in grains (pg fungal DNA ng barley DNA^−1^) and NFNB disease index (%) in Quench and Sunshine cultivars scored at BBCH 80. Three replicates for each experimental treatment in 2019 and 2020 are shown separately.

**Table 1 pathogens-11-00291-t001:** Grain yield (t ha^−1^), protein content (%), and kernel specific weight (kg hL^−1^) of the six tested barley cultivars following fungicide applications and in the untreated controls. In the table, 2019 and 2020 data are reported. The different letters on the MCP columns show statistically different values.

Cultivar	Treatment	Grain Yield 2019			Grain Yield 2020			Protein Content 2019			Protein Content 2020			Specific Weight 2019			Specific Weight 2020		
		t ha^−1^	SE^a^	MCP^b^	t ha^−1^	SE	MCP	%	SE	MCP	%	SE	MCP	kg hL^−1^	SE	MCP	kg hL^−1^	SE	MCP
**Quench**	Untreated control	10.5	0.10	b	7.7	0.33	a	10.0	0.26	a	12.1	0.40	a	66.0	0.06	b	67.4	0.15	a
	Aviator Xpro	12.0	0.10	a	8.1	0.24	a	9.9	0.32	a	12.5	0.46	a	68.6	0.32	a	67.4	0.20	a
	Elatus Era	12.1	0.21	a	8.1	0.19	a	10.2	0.20	a	11.4	0.09	a	68.1	0.03	a	67.4	0.12	a
	Priaxor	11.6	0.25	a	7.6	0.54	a	10.2	0.29	a	11.6	0.25	a	68.6	0.30	a	66.9	0.35	a
**Sunshine**	Untreated control	7.4	0.41	a	7.7	0.09	b	11.9	0.10	a	12.3	0.21	a	67.2	0.26	b	66.6	0.54	a
	Aviator Xpro	8.1	0.58	a	8.1	0.09	a	11.7	0.17	a	11.8	0.15	a	68.8	0.25	a	67.2	0.32	a
	Elatus Era	8.1	0.46	a	8.1	0.15	a	12.0	0.07	a	12.2	0.23	a	68.5	0.17	a	67.6	0.23	a
	Priaxor	8.0	0.15	a	8.3	0.07	a	11.6	0.21	a	11.9	0.15	a	68.7	0.54	a	66.8	0.12	a
**Atomo**	Untreated control	8.3	0.43	a	6.8	0.55	a	12.9	0.66	a	13.9	0.17	a	67.9	0.56	a	63.3	0.47	a
	Aviator Xpro	8.7	0.57	a	7.1	0.64	a	12.8	0.43	a	14.0	0.03	a	68.2	0.38	a	62.8	0.31	a
	Elatus Era	8.6	0.38	a	7.3	0.44	a	13.5	0.30	a	14.1	0.26	a	66.9	1.47	a	62.4	0.81	a
	Priaxor	9.1	0.21	a	6.9	0.69	a	12.6	0.54	a	14.2	0.15	a	68.1	0.62	a	63.2	0.94	a
**Capricorn**	Untreated control	10.7	0.36	b	8.4	0.07	a	11.9	0.10	a	12.5	0.03	a	64.2	0.21	a	63.6	0.10	a
	Aviator Xpro	11.5	0.18	a	8.2	0.25	a	11.7	0.17	a	12.1	0.20	a	65.6	0.82	a	63.2	0.31	a
	Elatus Era	11.9	0.17	a	8.5	0.06	a	12.0	0.07	a	11.9	0.29	a	66.7	0.32	a	62.6	0.28	a
	Priaxor	11.3	0.08	ab	8.1	0.23	a	11.6	0.21	a	11.8	0.52	a	65.7	0.73	a	63.2	0.18	a
**Ketos**	Untreated control	9.1	0.53	b	6.9	0.68	a	11.3	0.35	a	12.3	0.13	a	66.3	0.00	c	61.9	1.32	a
	Aviator Xpro	10.6	0.37	a	7.5	0.65	a	11.2	0.25	a	12.1	0.31	a	67.5	0.32	ab	62.1	1.41	a
	Elatus Era	10.9	0.05	a	7.7	0.53	a	11.5	0.22	a	12.0	0.17	a	68.0	0.18	a	63.3	1.24	a
	Priaxor	10.6	0.36	a	7.5	0.56	a	11.5	0.24	a	12.2	0.32	a	66.5	0.55	bc	61.9	0.95	a
**Alimini**	Untreated control	10.1	0.04	a	7.6	0.08	a	12.3	0.17	a	12.2	0.03	a	64.2	0.52	a	60.2	0.29	a
	Aviator Xpro	10.9	0.39	a	7.0	0.74	a	12.0	0.31	a	12.2	0.09	a	64.4	0.47	a	60.4	0.71	a
	Elatus Era	10.6	0.50	a	7.8	0.19	a	11.6	0.18	A	12.2	0.24	a	65.1	0.19	a	59.9	0.39	a
	Priaxor	10.6	0.18	a	7.6	0.09	a	11.9	0.18	A	12.1	0.12	a	64.4	0.44	a	60.5	0.66	a

^a^ SE = ± standard error; ^b^ MCP = multiple comparison procedure (Duncan’s multiple comparison test).

**Table 2 pathogens-11-00291-t002:** Characteristics of the barley cultivars analyzed in the study.

Cultivar	Row	Type of Grain	Height of Vegetation	Harvesting Time	Typical Italian Cultivation Area	Main Destination
Quench	Two	Covered	Low	Early	Centre	Malting
Sunshine	Two	Covered	Low	Medium	All	Malting
Atomo	Two	Covered	Medium	Early	All	Feed
Capricorn	Two	Covered	Medium/High	Medium/Late	All	Feed
Ketos	Six	Covered	High	Medium	All	Food/Feed
Alimini	Six	Covered	High	Medium	North	Feed

**Table 3 pathogens-11-00291-t003:** Fungicide treatments, active ingredient concentrations, and doses used in the experiments.

Treatment	Active Ingredient	Active Ingredients Concentration (g L^−1^)	Mode of Action	Application Time	Dose (L ha^−1^)
Untreated control	-	-	-	-	-
Aviator Xpro	prothioconazole + bixafen	150 + 75	DMI ^a^ + SDHI ^b^	BBCH 39	1
Elatus Era	benzovindiflupyr + prothioconazole	75 + 150	SDHI + DMI	BBCH 39	1
Priaxor	fluxapyroxad + pyraclostrobin	75 + 150	SDHI + QoI ^c^	BBCH 39	1.5

^a^ DMI = demethylation inhibitor; ^b^ SDHI = succinate dehydrogenase inhibitor; ^c^ QoI = quinone-outside inhibitor.

## Data Availability

Not applicable.

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
