# Peer review of "Management of Pyrenophora teres f. teres, the Causal Agent of Net Form Net Blotch of Barley, in A Two-Year Field Experiment in Central Italy"

_pathogens, 2022, doi:10.3390/pathogens11030291_

Round 1

Reviewer 1 Report

MDPI - Pathogens

Title: Integrated management of Pyrenophora teres f. teres, the causal agent of net form net blotch of barley, in a two-year field experiment in central Italy

The work presented by the authors concerns a two-year study of the occurrence and pressure of pathogens of the species Pyrenophora teres (mainly Pyrenophora teres f. teres) threatening barley cultivation in Italy. The authors presented the results of field experiments (in 2018/2019 and 2019/2020) that they performed in the areas of natural occurrence and pressure from the selected pathogen. Six selected barley genotypes for various purposes (malt, feed and food) were used in the study, as well as three selected commercially used fungicides with three different models of action. The authors thoroughly present the problem of protecting barley and other cereals against pathogens.

In my opinion, it is worth changing the title of the publication because only the influence of cultivars (only 6) and fungicides (only 3) on the occurrence of the net blotch of barley was examined. However, the severity of the occurrence also depends on other factors that have not been studied in the study.

In the introduction (chapter 1) they emphasize the importance of the threat posed by Pyrenophora teres f. teres. They also indicate the essence of crop protection through appropriate agrotechnics or the use of appropriately selected fungicides with an appropriately adjusted model of operation (QoI, SDHI and DMI). The authors also draw attention to the essence of the appropriate selection of varieties (genotypes) in barley cultivation with the desired insensitivity characteristics.

The second chapter describes the obtained results of the experiments over the course of two years. Section 2.1 describes the results of the pathogen collection and identification process, both microscopic and molecular. The subsection fully and concisely describes the microscopic identification process, as well as the authors provided clear photographic documentation of the acquired pathogen.

In my opinion, instead of writing “varieties” the word “cultivars” should be used (applies to the whole work)

Subsection 2.2 describes the results of measurements of meteorological conditions at the site of the experiments. The authors collected data on temperature, precipitation and air humidity. The authors very carefully analyze the conditions that prevailed during the experiments as well as presented graphically in the form of a chart. These data clearly show that the years of the study differed significantly in terms of weather conditions, which also influenced the incidence of the disease (this is given in the results). In my opinion, with such variation in years, it was worth carrying out the third year of research. Two-year studies, only in one location, with large variations in weather conditions between years, may lead to wrong conclusions.

Section 2.3 describes the results of observations of the occurrence and severity of disease symptoms caused by Pyrenophora teres f. teres in 30 days from the application of selected fungicides. The authors compare the obtained observation results between the selected plant genotypes, combinations and years. The graph presented in the subsection (Figure 3) is legible and easy to perceive, you can see the dependencies presented in it. Similarly to the presented table (Table 1), it is readable and accessible to the reader. In my opinion, however, data on the weight of a thousand grains are missing.

Section 2.4 deals with the pathogen DNA content in relation to plant DNA. The authors clearly describe the obtained PCR results. The presented diagram (Figure 4) is also legible and easy for the reader to perceive.

Section 2.5 describes the relationship between the occurrence of disease symptoms and the biomass of the pathogen in plant tissues. In the presented graph, the authors indicate the correlation between the above-mentioned factors in the form of the Pearson correlation broken down according to genotypic dependencies and dependencies in growing seasons.

Chapter 3 is a discussion in which the authors carefully compare the obtained results with the rich literature. They indicate the essence of their own research in the context of the protection of cereals against P. teres f. teres (net form, NFNB) and P. teres f. maculata (spot form, SFNB). The presented results are compared in detail with the reports of other authors, which indicates a good preparation for the topic under study.

However, some questions arise.

How can you explain that there was a lot of DNA of Pyrenophora teres in the grain of the Sunshine cultivar in 2020, since there were very few disease symptoms on barley leaves?

Why was there no DNA of Pyrenophora teres in the grain of other barley cultivars, although there were relatively many disease symptoms on leaf?

Chapter 4 is the methodology used in the research. The chapter is divided into 4 sub-chapters. The first subsection describes the place where the field experiments were conducted. The pattern of conducted experiments is also described. Moreover, the table (Table 2) lists the genotypes of barley plants used in the research along with their most important characteristics. It also includes all information on the plant protection products used, the method of their application and dosage (Table 3), as well as all information on other treatments.

The information provided shows that a large dose of the working liquid was applied (500 L ha-1). In practice, it is even lower, even 200-300. 500 L ha-1

Among other things, data on nitrogen fertilization have been provided. These data show that the doses of fertilization in the years were very different. In the first year of the study, there were 180 kg N ha-1, 120 kg N ha-1 in the second year. The much higher dose in the first year probably contributed to the increase in the severity of the disease, as is known from the literature. Unfortunately, the authors of the publication did not comment on this fact. It should also be included in the discussion, supported by publications.

The information provided shows that a large dose of the working liquid was applied (500 L ha-1). In practice, it is lower, usually even 200-300 L ha-1.

The subsection also contains information on how to observe plants in terms of the occurrence of plant symptoms and how to evaluate and assess the infestation of the tested plants. Unfortunately, the information concerning the assessment of the severity of the disease symptoms is laconic. Actually, it is not known how the assessment was carried out. The number of plants was not reported. The flag leaf "or" the leaf just below flag leaf was rated. It should not be like that. The same leaf should be judged as the intensity varies considerably from one leaf to another, especially in barley. The method of assessing the severity of the disease is not understandable for me. Why define "percentage of plants per plot showing net blotch symptoms" - it adds nothing. Why calculate the disease index (expressed in %), when it was possible to calculate the average percentage of leaf area with symptoms of disease (also expressed in %). Who is the author of the formula for calculating the disease index? Was the severity graded? If so, what do the various degrees mean? This information should be easy to understand.

In subsection 4.2, the authors describe the method and methods of isolation and identification of pathogens in the conducted research. The authors thoroughly and in detail describe the method of isolation and microscopic identification of the pathogens studied, as well as describe in great detail the methodology of molecular identification of the obtained pathogens. The authors go into a very detailed description of molecular methods at each stage of extraction or PCR reactions.

Was the streptomycin sulfate dose too high (0.16 g L-1). This could inhibit mycelium growth.

Section 4.3 describes the method of determining the DNA content of the pathogen against the DNA of plants. The authors describe the extraction process in great detail and describe the qPCR reaction in great detail.

Section 4.4 describes the statistical analyzes used by the authors. The obtained results were analyzed using one-way ANOVA. Each tested parameter was analyzed in a similar way. Additionally, the analysis of the Pearson correlation between the studied years and two genotypes in the context of the DNA content of the pathogen against the background of the plant DNA and the occurrence and intensity of disease symptoms was used.

In summary, the authors undertook the task of protecting barley crops against Pyrenophora teres f. teres in Italy. Considering the importance of the problem of losses caused by the pathogens studied and the essence of barley cultivation as an important plant in the food industry, the research undertaken seems to be very much needed. Nevertheless, the work has a few flaws(most of them of minor importance) that should be mentioned.

  • In Figure 2, the red frames could be better fitted
  • In Figure 5, the authors describe the correlation between the studied variables, which are the pg DNA content of the pathogen / ng plant DNA, and the occurrence of disease symptoms. The chart describes the significance of the correlation for and the fit factor. Nevertheless, data for only two genotypes out of the six subjects were used. Only a part of the data was used, without indicating the relationship for the remaining plant genotypes. Even if they were irrelevant, any mention would be good. At this point, it seems that the data "that best fit the research hypothesis" have been used.
  • In the materials and methods chapter, the authors presented the equation they used to calculate the disease index on plants. However, it was not mentioned according to whom. No citation is provided.

Misspellings of the names of the perpetrators of diseases:

It is “Helmintosporium”, and it should be “Helminthosporium”

It is “Rhyncosporium”, and it should be “Rhynchosporium”

Another

It is “fluroxypyr144”, and it should be “fluroxypyr 144”

Pyrenophora teres f. teres is not seed-born pathogen only. There may also be other sources of infection.

In the summary of the paper, too much generalization was used, not supported by research results. Omitting certain data leads to wrong conclusions.

Author Response

Dear reviewer 1, 

please see the point-by-point response in the attachment. 

Thanks. 

Reviewer 2 Report

The authors describe the occurrence of net blotch on barley in a two-year field experiment.

The introduction is well written, the results are clearly presented, the experimental design and the methods are well chosen and comprehensible.

Figure 1: a) and b) in the figure are hardly recognizable. Please change the color or the background

Figure 2 and Figure 5: Please use a bigger font size (min. 10) for the captions and the labels of the axes.

Figure 4: no difference in variety Sunshine between pathogen biomass in the grains harvested in both years. Do you have any suggestions why Quench showed differences while Sunshine did not and while it was not detectable in the other varieties? Please elaborate it in a paragraph in the discussion

85: the present study shows the results

108 – Change the sentence to: After amplification of the ITS region from the extracted DNA originating from the monoconidial colony developed on PDA, using primer ITS1 and ITS4, the isolated pathogen was identified, by BLAST analysis, as P. teres.

211: species-specific primers

in line 214 and 225 I assume you are referring to figure 4 and not figure 3. Please correct it.

332: effect of fungicide treatments

Author Response

Dear reviewer 2, 

please find the point-by-point response in the attachment. 

Thanks. 
